# Stakeholder Opinions on the Issues of the Central Arizona Groundwater Replenishment District and Policy Alternatives

Rebecca F. A. Bernat [1,*], Sharon B. Megdal [2], Susanna Eden [2] and Laura A. Bakkensen [3]

1 Department of Environmental Science, College of Agriculture and Life Sciences, The University of Arizona, Tucson, AZ 85721, USA
2 Water Resources Research Center, The University of Arizona, Tucson, AZ 85719, USA
3 School of Government and Public Policy, College of Social and Behavioral Sciences, The University of Arizona, Tucson, AZ 85721, USA
* Correspondence: rebeccabernat@arizona.edu

**Abstract:** Arizona has been at the forefront of groundwater management since the establishment of the Groundwater Management Act in 1980. The Central Arizona Groundwater Replenishment District (CAGRD) is a groundwater management mechanism that facilitates development in regions of Central Arizona where the use of groundwater is limited by law. Several stakeholders have raised concerns about some of the CAGRD's operations; however, stakeholders have yet to agree on the definition of the problems, let alone how the CAGRD might be improved. This study uses statistical and inductive thematic content analysis of a survey to determine (1) the CAGRD issues that stakeholders view as problems and (2) whether opinions differ significantly among different stakeholder groups. This study also uses deductive thematic content analysis to examine semi-structured interviews with CAGRD experts in order to find potential solutions to the CAGRD-related issues that are considered problems by stakeholders. The survey results show that long-term uncertainties related to the availability of renewable water supplies and hydrologic disconnect, where groundwater pumping and replenishment take place in different sub-basins, are stakeholders' chief concerns. Sector affiliation and CAGRD membership status are associated with stakeholders' opinions on some, but not all, questions. The potential policy changes offered address problems identified by stakeholders. This research will inform forthcoming policy discussions regarding groundwater management in Central Arizona as the state's decision makers look to improve the CAGRD in the context of water scarcity exacerbated by climate change.

**Keywords:** policy analysis; stakeholders; content analysis; interviews; Central Arizona Groundwater Replenishment District; water management; water policy; replenishment; managed aquifer recharge





## 1. Introduction

Arid regions of the world are facing unprecedented challenges to water supplies for growing populations. The State of Arizona in the Southwestern United States has growing water demands, significant groundwater overdraft, and surface water supplies with diminishing reliability [1]. The study described herein uses Arizona as a relevant case study supporting efforts to improve an existing groundwater management tool intended to protect against groundwater depletion while allowing groundwater-dependent growth. By examining stakeholder opinions regarding the Central Arizona Groundwater Replenishment District (CAGRD), this study identifies the chief problems with this entity and potential solutions, as indicated by stakeholders through a survey and interviews.

In 1980, the Groundwater Management Act created Active Management Areas (AMAs) in which groundwater use is regulated by the Arizona Department of Water Resources (ADWR). Within an AMA, development depends on the demonstration of a 100-year Assured Water Supply (AWS) [2]. The term development (also called real-estate development or property development) refers to the activities of the development community for the

production of residential and commercial properties. The development community encompasses developers, landowners, and homebuilders. To sell or build on lots in an AMA, the development must hold an AWS certificate or be located within the service area of a water supplier with an AWS designation issued by the ADWR. One of the criteria for an AWS certificate or designation is consistency with the groundwater management goal of the AMA. This requirement limits the amount of groundwater that water providers may pump or that homeowners may use in an AMA. The remainder of the supply must be renewable. Under Arizona law, groundwater is considered nonrenewable.

The passage of the Groundwater Management Act was one of the conditions for federal funding to build the Central Arizona Project (CAP), a 336-mile canal system completed in 1992 that transports water from the Colorado River on the western boundary of Arizona uphill to Central Arizona. In 1971, the Arizona legislature created the Central Arizona Water Conservation District, a governmental subdivision of Arizona encompassing Maricopa, Pima, and Pinal Counties, to operate the CAP. Generally, Colorado River water delivered via the CAP canal is referred to as CAP water.

The Groundwater Management Act required the establishment of an AWS program, which took the form of interim rules until 1995 [3]. The AWS Rules, adopted in 1995, are the cornerstone of the Groundwater Management Act [4,5] because they are essential to limiting increases in groundwater use in AMAs. Since not all developers and water providers can access sufficient renewable water to meet AWS requirements, the Arizona State legislature established the CAGRD in 1993, prior to the adoption of the AWS Rules. Operated by the Central Arizona Water Conservation District, the CAGRD provides a mechanism for developers and municipal water providers to show conformity with the management goals of the Central Arizona AMAs. Membership in the CAGRD supports an application for an AWS certificate or designation because the CAGRD replenishes a portion of the groundwater pumped by its members. Replenishment is a mechanism that consists of replacing groundwater withdrawals with injection or infiltration of water considered renewable, such as CAP water or treated wastewater. Replenishment rules are established under the Underground Water Storage, Savings and Replenishment Act and other statutes [6–9].

The CAGRD has been a crucial mechanism to allow for economic development in the study area (Figure 1). By becoming CAGRD members, developers and municipal water providers without access to enough renewable water can demonstrate an AWS, and the CAGRD is obligated to replenish the excess groundwater members use. The AWS Rules require the demonstration of physically, legally, and continuously available water supplies for 100 years; financial capability to treat and deliver water that meets water quality standards; consistency with the periodic management plans required for each AMA; and consistency with the statutory groundwater management goal of the AMA [4,5]. Broadly defined, the overarching goal of the Phoenix, Tucson, and Pinal AMAs is sustainable groundwater use. Consistency with the management goals, and thus adherence to the AWS Rules, limits the volume of groundwater AWS applicants can use.

Municipal water providers that want to facilitate development in their service areas must apply for a designation of AWS. Designated municipal water providers may use the volume of groundwater that the ADWR calculates is consistent with the AMA's management goal. If the municipal provider anticipates a need for more groundwater than the AWS rules permit, they must enroll in the CAGRD as a member service area (MSA). The CAGRD will replenish the excess groundwater pumped by the MSA, as reported annually. Developers of land not located within the service area of a designated municipal water provider must apply for a certificate of AWS before building or selling lots in a subdivision of six or more lots [2]. If developers anticipate the need for more groundwater to obtain a certificate of AWS, they may enroll the subdivision as a CAGRD member land (ML). In exchange, the CAGRD replenishes the excess groundwater pumped for use by the ML.

Several issues related to the CAGRD have gained increasing attention among scholars and stakeholders [10,11], who have raised concerns about the CAGRD and suggested solu-

tions to some of these concerns. However, all stakeholders are unlikely to agree on which issues are problems, and no systematic polling of stakeholders has been attempted before. For the purposes of this analysis, stakeholders are defined as individuals or entities that affect, are affected by, or have an interest in the outcomes of decisions and actions [12–15].

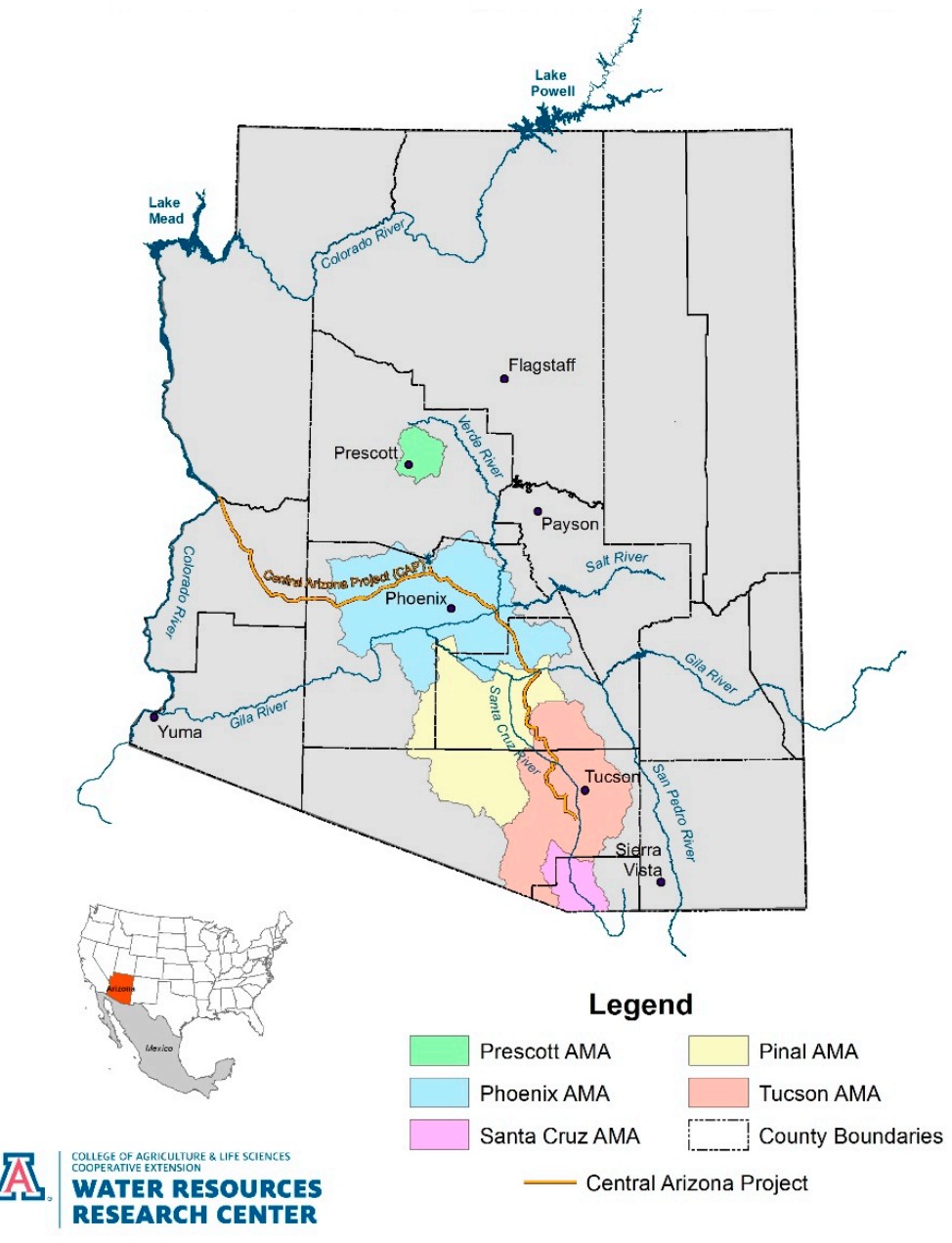

**Figure 1.** Map of Arizona showing the AMAs and county boundaries.

The role of stakeholders in water resources management is critical. Kujala et al. [16] conducted a worldwide literature review of 90 articles and discussed theories and concepts related to the role of stakeholders in general. Many studies about water resources management in Canada [17], China [18], Denmark [19], Brazil [15], South Africa [20], Spain [21,22], the United States [23,24], and other regions [25,26] highlight the importance of stakeholder engagement. After reviewing 20 peer-reviewed papers collected in a Special Issue of Water, Megdal et al. [27] not only concluded with the "importance of stakeholder engagement in water governance issues", but also that there is "an essential need for scientific publication outlets that present varieties of case studies and discuss best practices". In Arizona

specifically, Eden et al. [28] found that "stakeholders provided critical input to hydrologic modeling analyses".

The novelty of this article lies in the fact that, unlike the aforementioned studies that analyzed the effectiveness of stakeholder processes and the associated best practices, it shows how stakeholder feedback is used in policy analysis. This study takes the state of the art one step further by actually analyzing the opinions on water issues provided by stakeholders. Most stakeholders surveyed and interviewed in this study had already been engaged by a committee established by the Arizona's Governor Office. In terms of innovation, this study shows how stakeholders can contribute to drafting better policies: stakeholder survey answers are used to identify timely policy problems; CAGRD expert opinions are used to find solutions to the identified problems.

In 2007, Avery et al. identified three main issues regarding CAGRD operations: reliance in part on short-term supplies, a growing membership that was increasing replenishment obligations, and a "hydrologic disconnect" between the area from which members pump groundwater and the location of CAGRD replenishment. This disconnect does not occur in all cases of CAGRD replenishment.

Before 2004, the CAGRD relied on surplus water for recharge, called excess CAP water, to fulfill its replenishment obligations. In its 2005 Plan of Operation, the CAGRD "identified a significant gap between its water supplies and replenishment obligation" [29] and established a goal of 50% long-term supplies and 50% short-term supplies for their acquisition plan [30]. In 2014, the CAGRD planned to fulfill its obligations with the acquisition of effluent, long-term storage credits (water already stored in aquifers) and higher-priority CAP water, rather than excess CAP water [31].

By law, the CAGRD may not refuse to accept new members, and its replenishment obligation will increase as new entities join. The CAGRD projects an increase in its replenishment obligation of 33% between 2015 and 2023 [32]. Moreover, the CAGRD must look for new sources of supply to replace the excess CAP water that is no longer available. Therefore, the CAGRD must look for alternative sources of supply.

Another issue concerning observers is the aforementioned "hydrologic disconnect". Currently, CAGRD activities may allow aquifer depletion in areas where developments pump groundwater from one sub-basin while the CAGRD replenishes in a separate and possibly distant sub-basin. This can occur legally if both sub-basins are within the same AMA [33]. In addition, in the Phoenix AMA the CAGRD is required to replenish in the same division of the AMA (east or west) as the pumping occurs, but only "to the extent reasonably feasible" [34]. This hydrologic disconnect can have adverse effects on the aquifer system in the long term [1,10,35]. Moreover, the hydrologic disconnect might compromise the future ability of water users to pump groundwater or even to recover water stored underground. The Underground Water Storage, Savings, and Replenishment Program allows water recovery to occur only within the area of impact of the storage facility or in an area where the average annual rate of decline is less than four feet per year [36–38]. Groundwater pumping by CAGRD members is not limited by these regulations.

Different CAGRD issues have been raised [39–43] by over 110 stakeholders between 2019 and 2022 [44]. The first one related to a hydrologic disconnect, which happens when there is a physical disconnect between the area where water is added to the aquifer and the area where it is extracted [41]. This issue is not unique to the CAGRD, as it concerns not only activities related to CAGRD replenishment, but also activities related to recharge. Although AMA boundaries were drawn mainly to coincide with major aquifer boundaries, each AMA aquifer is divided into sub-basins, and pumping water into one sub-basin while replenishing or recharging water in another may lead to areas of rising water tables unconnected with areas of water-table declines. The second issue related to the AWS Program, which permits some residential developments and municipalities in Central Arizona AMAs to rely primarily on groundwater if groundwater is replenished. This practice raises concerns about groundwater sustainability and the risk of localized groundwater depletion. Groundwater depletion has consequences for future growth

because water levels may fall below regulatory limits, which may prevent AWS applicants from showing the physical availability of groundwater: to use groundwater, it must be physically available for 100 years of demand by the development. Pumping may not lower water tables beyond 1000 feet below the land surface in Phoenix and Tucson AMAs or 1100 feet below the land surface in the Pinal AMA [45]. Furthermore, allowing water users to rely on groundwater acts as a disincentive to acquiring alternative supplies. Other issues were raised about the CAGRD [41]: the long-term uncertainties associated with the availability of renewable water supplies to meet the CAGRD's replenishment obligation, the CAGRD's method of charging for ML replenishment services, and the statutory framework for ADWR review of the CAGRD Plan of Operation.

Creating the CAGRD was essential to the passage of the AWS Rules, the cornerstone of the Groundwater Management Act. The lack of agreement among stakeholders about the CAGRD prompted this analysis, which addresses the following research questions: What issues do stakeholders perceive as problems? What do they think are the most important problems to address? How do opinions compare across categories of stakeholders? What solutions could be implemented to address the CAGRD issues that stakeholders agreed were problems?

We differentiate the word "issue" from the word "problem". An "issue" is a topic under discussion that needs debating, whereas a "problem" requires solving. In a nutshell, a consensus must be reached for an issue to be declared a problem. This distinction is important because problem structuring (i.e., producing information about the problem to solve) is the basis of policy analysis [46,47]. Finding the right problem(s) is essential in order to avoid solving the wrong problem, which is a common mistake in policy analysis [46].

The first objective of this study was to recognize stakeholder opinions in order to solve the right CAGRD policy problems. Indeed, before searching for solutions, it is necessary to structure the problem related to the CAGRD by locating points of consensus around which to build policy alternatives. In light of stakeholder disagreement about what constitutes a CAGRD problem, this study first assessed stakeholders' opinions on CAGRD issues. The second objective of this study was to assess the solutions shared by CAGRD experts in order to consider policy alternatives for the issues deemed problematic by the majority of stakeholders. Using statistical, deductive, and inductive thematic content analysis, this study presents an innovative combination of methods designed to conduct policy analysis.

## 2. Materials and Methods

### 2.1. Survey Design

#### 2.1.1. Data Collection Method

We designed a survey to elicit information about stakeholder opinions surrounding the CAGRD as surveys are quick and straightforward ways to obtain information about people's attitudes and opinions [48]. This survey was web-based, allowing participants to complete it in their own time, and it typically took less than 10 min to complete. We chose Qualtrics as the digital survey tool and the data collection method.

The survey produced quantitative and qualitative data [49] in answer to three stakeholder identification questions (mandatory), six opinion questions with comments (optional), and one ranking question (optional). The information collected in the survey could be compared across stakeholders because all the individuals surveyed answered the same questions [50]. The population of this survey comprised entities that affect, are affected by, or have an interest in the outcomes of decisions and actions regarding the CAGRD. The questions in this survey were carefully designed to ensure that the differences in answers could be associated with differences in stakeholders characteristics [50].

The first three questions collected basic information to accustom the respondents to the survey. We asked stakeholders to select the professional category, sector, and county that best described them for the purpose of the study [51]. Then we asked respondents for their opinions on six questions, with the possibility of commenting after each question. Since survey respondents tend to agree with statements more often than disagree, we chose to

formulate "what do you think of . . . "-type questions. This limited acquiescence response bias by avoiding leading statements [52]. Answer options to opinion questions were: "This is a problem", "This is not a problem", "I don't know", and "No opinion". The answer "I don't know" was for respondents who believed they had inadequate knowledge about the issue [50]. The answer "No opinion" was for respondents who had not formed an opinion on the topic or did not wish to share it. We chose this format, rather than ordinal scales, such as the Likert scale, because these may provide unreliable measurements, as responses are relative to the state of the stakeholder and produce too much variation [50]. Ordinal scales also have a higher cognitive burden [53].

Finally, we asked stakeholders to rank issues. The opinion and ranking questions were all optional to avoid the probability of collecting incorrect information from unwilling respondents.

To obtain nuanced answers, we offered respondents the opportunity to comment after each of the six opinion questions. Open questions have the advantage of eliciting unanticipated answers; they also allow individuals to respond according to their own definitions of issues; thus, they reduce respondents' frustration and survey dropout [50]. While the value of many open-ended questions may be limited by vague and incomplete answers [54], they produce complementary data that increase the value of constrained choice data.

### 2.1.2. Dependent Variable: Selection of Issues to Test

To establish which CAGRD issues are perceived as problems among Arizona's stakeholders and to ascertain the most critical problems that need attention, we conducted a stakeholder opinion survey about six CAGRD-related issues discussed by a committee established by the Arizona's Governor Office:

*Opinion question 1: What do you think of recharging water (at underground storage facilities or groundwater savings facilities) and recovering it in hydrologically disconnected areas in Central Arizona AMAs?*
*Opinion question 2: What do you think of replenishing pumped groundwater in hydrologically disconnected areas?*
*Opinion question 3: What do you think of the fact that the Assured Water Supply program permits some residential developments and municipalities in Central Arizona AMAs to rely primarily on groundwater?*
*Opinion question 4: What do you think of the long-term uncertainties related to the availability of renewable water supplies to meet CAGRD's replenishment obligation?*
*Opinion question 5: What do you think of CAGRD's method of charging member land homeowners for replenishment services?*
*Opinion question 6: What do you think of the statutory framework for ADWR review of the CAGRD Plan of Operation?*

We assumed that the respondents knew all the words and concepts in each question. Each question generated a categorical opinion variable (see Appendix A, Table A1). Categorical questions are adequate to obtain easy-to-analyze counts and percentages [52]. To determine which problems require immediate attention, we asked respondents to rank issues by order of priority (see Appendix A, Table A2). Answers were optional and randomly displayed to eliminate order bias. Respondents could choose to only rank a subset of issues.

### 2.1.3. Stakeholder Identification Sampling

Effective stakeholder engagement is required [55] to facilitate dialogue related to decision-making processes [15]. Stakeholder identification is an essential step in stakeholder analysis [14]. We listed CAGRD experts, as well as individuals and organizations discussed in the CAGRD literature [10,11,35,56]; lawyers representing/who had represented developers or homebuilders; and other well-known CAGRD experts, such as consultants and academic researchers. These experts were identified by the authors because of their

known experience in the field. To ensure that the population sample was representative of all stakeholders who have been publicly engaged in the CAGRD, we added individuals who commented during public CAGRD meetings [57,58].

2.1.4. Independent Variables: Stakeholder Categorization by Group, Sector, and CAGRD Membership

To determine how opinions compared among stakeholders, we asked the respondents to select the primary category of stakeholders they represented or most closely identified with. The literature and observations informed these categories [5,55,59–62]. Knowing how opinions may differ between the public and private sectors may help gauge which groups are likely to support policy change.

To determine how opinions compared among different sectors, we asked respondents whether they identified as a private company, a public entity, a nonprofit, or as another type of organization. To determine how opinions compared between CAGRD members and non-members, we asked respondents whether or not they worked for a CAGRD MSA or owned property in a CAGRD ML. Appendix B displays the complete list of questions asked to collect information about the independent variables.

*2.2. Interviews with CAGRD Experts*

In order to gain information to develop policy alternatives that could be used to improve CAGRD operations, we requested interviews with 35 CAGRD experts. These experts were selected based on the authors' experience in the field, and using snowball sampling, beginning with individuals known to have expertise. We conducted semi-structured interviews with the 16 CAGRD experts who agreed to be interviewed. Interviewees were asked to define the advantages and disadvantages of the way the CAGRD operates, to define issues, and then offer potential solutions. Interviewees were also invited to share any other information about the CAGRD. Appendix C displays the list of questions asked the CAGRD experts during the interviews.

*2.3. Data Acquisition, Processing, and Analysis*

2.3.1. Central Arizona Groundwater Replenishment District Issues

Survey responses were collected from 10 August 2021 to 1 September 2021. The survey was shared with 137 individuals via email. Respondents were sent a link to the online Qualtrics survey form where they recorded their answers, and the data were exported to Excel for analysis. We used descriptive statistics to determine the percentages of stakeholders that perceived each issue as a problem and which issues were the top priorities.

We used a method of qualitative content analysis to analyze comments made by respondents on each CAGRD issue. Specifically, we used a thematic analysis with an inductive approach [63] by highlighting sections of the text and assigning a topic to them. Out of the 41 respondents, 30 left at least one comment while taking the survey. In total, 134 comments were collected. For each opinion question, topics mentioned by at least two individuals were recorded. This resulted in 45 comments selected for the thematic analysis. Comments that reaffirmed the questions without additional content were not included.

2.3.2. Chi-Square Test to Test Hypotheses about Stakeholder Differences in Opinions

To analyze whether stakeholder opinions differed across professional categories, sectors, and membership statuses, we used a Chi-square test of independence. The objective of a Chi-square test is to determine whether there is a significant association between dependent and independent variables or whether an association is random. The independent variables in this study were nominal variables (e.g., public entities, private companies, and nonprofit organizations). We chose a Chi-square test because it is suitable to conduct analyses with more than two groups of nominal data [64].

We created a cross-tabulation for each of the six questions that asked for opinions on potential problems, and then tested the responses for different groups of stakeholders:

category (Hypothesis 1), sector (Hypothesis 2), and CAGRD membership status (Hypothesis 3). In total, we conducted 18 Chi-square tests. The null hypothesis ($H_0$) was that the proportion of opinion responses from each subgroup would be the same. The alternative hypothesis ($H_A$) was that the distribution of their opinions would show statistically significant differences.

**Hypothesis 1 ($H_0$):** $P_{consultant\ \&\ real\ estate} = P_{governmental\ entity} = P_{municipal\ water\ provider} = P_{NGO} = P_{other}$;

**Hypothesis 1 ($H_A$):** $P_{consultant\ \&\ real\ estate} \neq P_{governmental\ entity} \neq P_{municipal\ water\ provider} \neq P_{NGO} \neq P_{other}$.

**Hypothesis 2 ($H_0$):** $P_{private\ company} = P_{public\ entity} = P_{non\text{-}profit}$;

**Hypothesis 2 ($H_A$):** $P_{private\ company} \neq P_{public\ entity} \neq P_{non\text{-}profit}$.

**Hypothesis 3 ($H_0$):** $P_{CAGRD\ member} = P_{non\text{-}CAGRD\ member}$;

**Hypothesis 3 ($H_A$):** $P_{CAGRD\ member} \neq P_{non\text{-}CAGRD\ member}$.

*In the above, P = proportion of responses from a subgroup of stakeholders.*

2.3.3. Interview Data Collection

The 16 interviews ranged in duration from 30 to 90 min, yielding about 150 pages of transcripts. We used a method of qualitative content analysis to identify and describe the solutions offered by the experts. Specifically, we proceeded with a thematic analysis with a deductive approach [63] by highlighting sections of the transcripts that provided solutions to CAGRD activities. Solutions mentioned by at least two experts were recorded.

**3. Results**

*3.1. Representation of Stakeholders*

Out of the 137 individuals surveyed, 41 responded. The majority of respondents were in Maricopa County (56%), while the rest were in Pima County (17%), Pinal County (7.3%), and Mohave County or a combination of these and non-specified counties (18%). Most stakeholders were located in the Maricopa, Pima, and Pinal Counties because the CAGRD operates in this geographic area. According to the 2020 US Census, 75% of Arizona's population was in Maricopa County [65], 18% was in Pima County [66], and 7% was in Pinal County [67]. Therefore, we can say that the responses generated were a valid representation of CAGRD stakeholders statewide because they mirrored the population across Central Arizona and outlying counties.

Overall, the distribution of respondents represented well the distribution of the sample survey (see Appendix D, Table A5), except that a larger percentage of private-sector consultants responded and none of the seven Native Nations surveyed responded (Native Nations water use is not regulated by the State). Several adjustments were made regarding the classificatory scheme based on the responses received. For purposes of the analysis (to increase precision and improve the relevance of the statistical tests), we reorganized the professional categories. We grouped consultants and real-estate developers in a category designated "Consultant and real estate" because the single member of the original "real estate" category also self-identified as a consultant. Additionally, we added the one academic respondent with the industry respondents to the category "Other". Finally, we moved five respondents to the professional NGO category when they either selected NGO as their second professional category of choice or they selected nonprofit as their sector. This reassignment increased the number of responses in the NGO category. These adjustments reduced the degrees of freedom and improved the relevance of the statistical tests.

After adjusting the categories, we found that the respondents represented mostly municipal water providers (27%), governmental entities (25%), NGOs (20%), and consultants and real estate (17%). The remaining respondents represented or identified with other categories (industry, academics, lawyers). Fifty-four percent of respondents identified with or represented the public sector, 26% identified as coming from the private sector, and 20% as coming from the nonprofit sector. Finally, only nine respondents were or represented a CAGRD MSA, and four represented MLs.

*Professional categories*. We conducted a Chi-square test for each of the six opinion questions for the three classifications of stakeholder groups (Table 1). The results showed that there was no significant difference between the ways all categories answered ($p > 0.05$). In other words, there was no relationship between the categories of stakeholders and their answers. Indeed, every test for every question accepted the null hypothesis at the $p > 0.05$ level. This means that the distribution of responses across professional categories was random.

**Table 1.** Results of the Chi-square test analysis at the $p < 0.05$ level. Rejecting the null hypothesis means there was a significant difference of opinion across groups of stakeholders.

| Group | Opinion Question 1 | Opinion Question 2 | Opinion Question 3 | Opinion Question 4 | Opinion Question 5 | Opinion Question 6 |
|---|---|---|---|---|---|---|
| Professional Category | Accept | Accept | Accept | Accept | Accept | Accept |
| Sector | Accept | Accept | **Reject** | **Reject** | **Reject** | Accept |
| CAGRD Membership | Accept | **Reject** | Accept | Accept | Accept | Accept |

*Sector categories.* The results showed that there was a relationship between the sectors of stakeholders and the answers to Questions 3, 4, and 5 at the $p < 0.05$ level. Therefore, the stakeholders from the private, public, and nonprofit sectors answered differently on questions 3, 4, and 5. Indeed, for sectors, we rejected the null hypothesis for Questions 3, 4, and 5 at the $p < 0.05$ level; we accepted the null hypothesis for Questions 1, 2, and 6.

*CAGRD membership categories.* The results showed that there was a relationship between the membership statuses of stakeholders and the answers to Question 2 at the $p < 0.05$ level. This means that CAGRD members and non-members answered differently on question 2. We therefore rejected the null hypothesis for Question 2 at the $p < 0.05$ level; we accepted the null hypothesis for Questions 1, 3, 4, 5, and 6 for membership status.

### 3.2. Identification of Significant Problems to Address

This section shows how the stakeholders prioritized the issues and the number of stakeholders who found each of the issues problematic. According to the descriptive data analysis of the survey's responses, the stakeholders ranked the importance of issues as follows: (1) the long-term uncertainties related to the availability of renewable water supplies to meet CAGRD's replenishment obligations, (2) the hydrologic disconnect caused by recharge and recovery, (3) the hydrologic disconnect caused by replenishment, (4) the AWS program permitting some residential development and municipalities in Central Arizona to rely primarily on groundwater, (5) the method of charging ML homeowners for replenishment services, and (6) the statutory framework for ADWR review of the CAGRD Plan of Operation. The answers to questions about individual issues supported this ranking. Appendix E summarizes the topics and arguments mentioned by stakeholders for each issue, which we discuss in each of the subsections.

### 3.2.1. First-Ranked Issue: Long-Term Uncertainties Related to the Availability of Renewable Water Supplies to Meet CAGRD Replenishment Obligations

A large majority of respondents (70%) were concerned about the long-term uncertainties related to the availability of renewable water supplies to meet the CAGRD's replenishment obligations (Figure 2). This issue was ranked first out of six as a problem

for stakeholders. The results from the statistical analysis showed that there was a strong difference of opinion among the stakeholders according to sectors. Indeed, 86% of public entities and 75% of respondents from the nonprofit sector agreed that this issue was a problem (Figure 3). By contrast, the private sector opinions were mixed (20% agreed and 50% disagreed) (Figure 3).

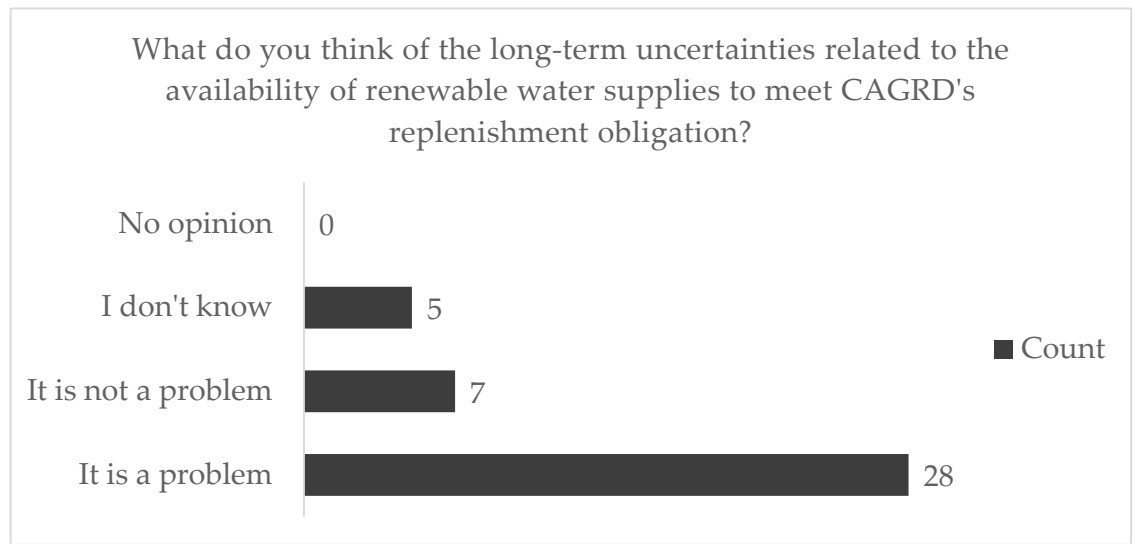

**Figure 2.** Survey answers to opinion question 4.

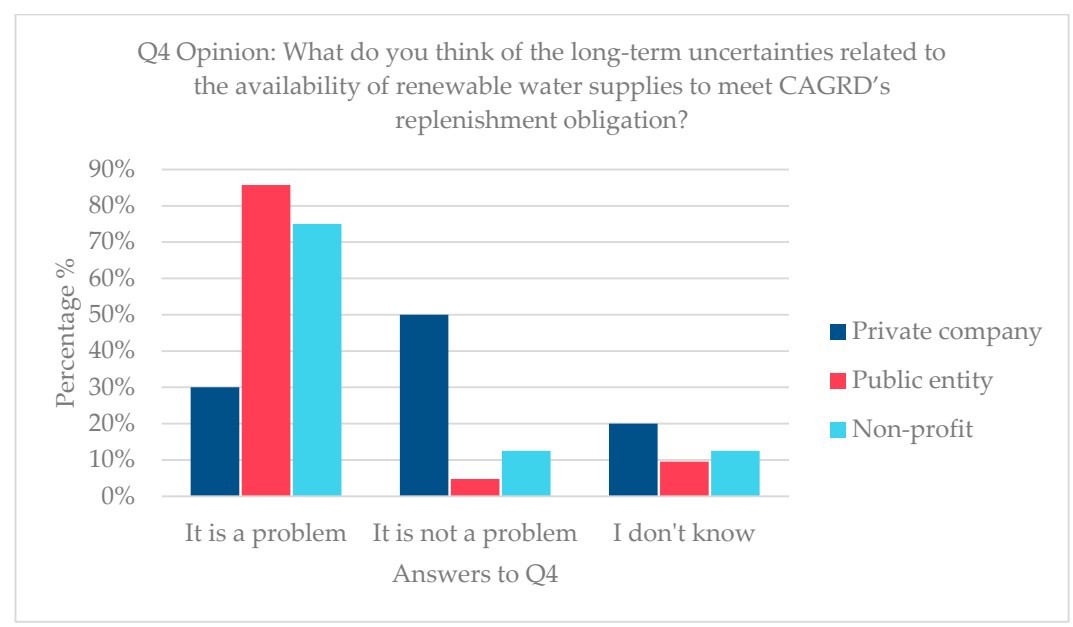

**Figure 3.** Distribution of answers given by each sector to opinion question 4.

Differences in opinion were supported by the analysis of the comments made by stakeholders. Indeed, of the six survey topics, stakeholders had the most to say about the availability of renewable supplies. Seven individuals mentioned the scarcity of renewable water and shortages on the Colorado River. Two stakeholders wrote about the competition to acquire water supplies and claimed that the CAGRD has an unfair advantage over municipal water providers due to its ability to purchase large volumes of water. While the cost of renewable water is expected to increase because of competition, one individual nuanced this statement, saying that the water remains inexpensive in Arizona compared to

other states. Additionally, five stakeholders said that augmenting supplies would mean the necessary development of the direct potable reuse of effluent or desalination of seawater. Lastly, as with the other issues, the idea that the problem is in fact beyond the administrative capacity of the CAGRD was brought up.

We therefore can confidently conclude that this issue is perceived as a problem by stakeholders, while opinions on the solution vary. It is important to note, however, that private entities may not support attempts to address this issue.

3.2.2. Second- and Third-Ranked Issues: The Hydrologic Disconnect

A large majority of respondents (77%) believed that recharging water and recovering it in hydrologically disconnected areas in Central Arizona AMAs was a problem (Figure 4), while a smaller majority (62%) believed that CAGRD replenishment was a problem specifically when it was disconnected from the pumped aquifer (Figure 5).

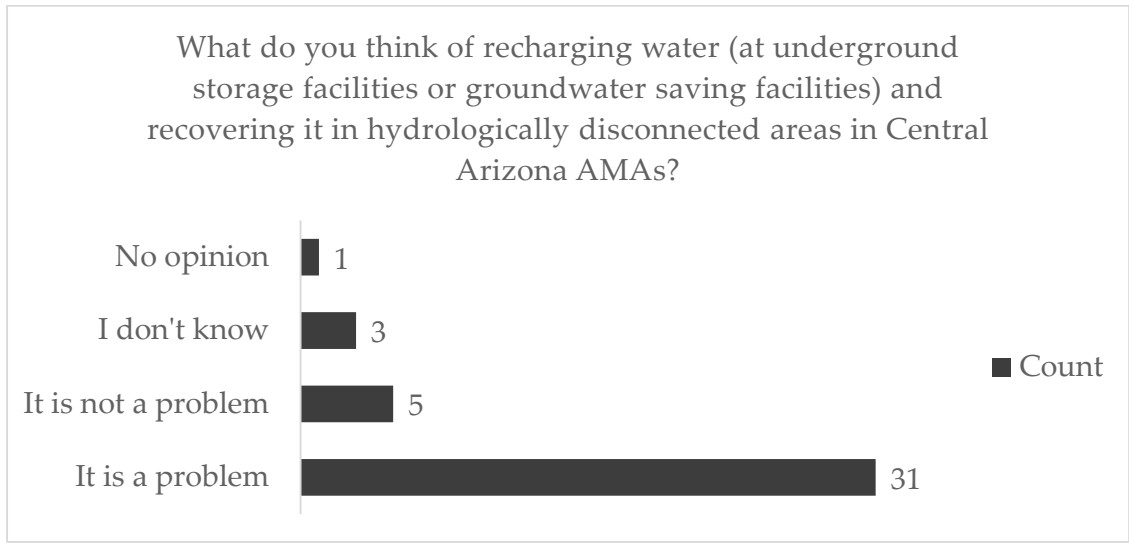

**Figure 4.** Survey answers to opinion question 1.

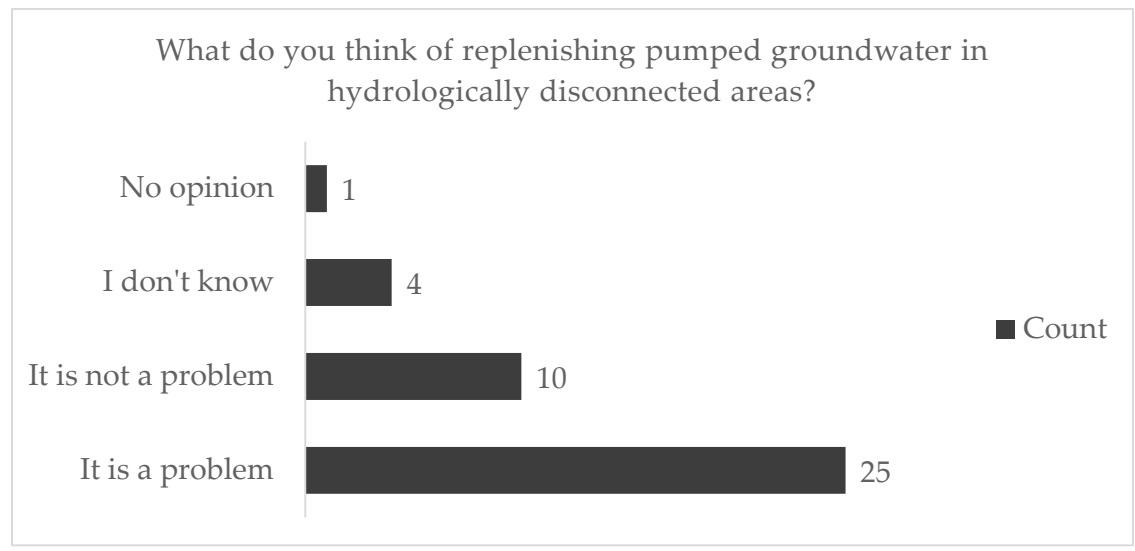

**Figure 5.** Survey answers to opinion question 2.

In the case of the hydrologic disconnect created by water recharge and recovery, the professional categories, sectors, and membership statuses of stakeholders were not found

to be statistically correlated with their responses. This indicates that there is a consensus across stakeholder categories on the fact that this issue is a problem. Specifically, seven respondents claimed that a hydrologic disconnect between the area of impact and the area of recovery creates groundwater overdraft or depletion in areas where water is not recharged. In essence, they are concerned that this procedure depletes aquifers. Some added that it can lead to groundwater declines, subsidence, fissuring, aquifer compaction, storage-capacity loss, and water-quality concerns. "I've seen fissuring and subsidence in the west valley with my own eyes", commented one respondent. Another opinion shared was that if disconnected recovery can potentially impact the water supplies stored by others, then aquifers do not benefit from recharge. As for the explanations of stakeholders who believed this was not a problem, for them the hydrologic disconnect was not a "one-size fits all issue" because it can be beneficial to recharge in disconnected areas with declining water tables. Additionally, stakeholders pointed out that the impact of the hydrologic disconnect is "vastly overstated" in comparison to the groundwater-level impacts of unreplenished groundwater pumping. The term "unreplenished groundwater" refers to groundwater that is "legally withdrawn without requirement or obligation to artificially replenish or replace that volume of water back into the aquifer" [68].

In the case of the hydrologic disconnect between pumping and replenishment, there was a difference of opinion between CAGRD members and non-members (at the 0.05 level). Almost 75% of non-CAGRD members believed it was a problem, while CAGRD members had mixed opinions (Figure 6). Since there were twice as many non-members surveyed, the conclusion that this issue is a problem is not supported with respect to the CAGRD member population. If this problem were to be addressed, the results of the survey could not help to determine definitively whether CAGRD members would support a policy change. Moreover, stakeholders blamed other water users for the hydrologic disconnect and said that the CAGRD's contribution is in fact negligible. They said that it can be an issue in the present, depending on the location, or that it could result in a problem in the long term (such as 100 years). They added that this discussion merits more analysis and that, as with the hydrologic disconnect associated with recharge and recovery, recharge may be beneficial in disconnected areas that need replenishment.

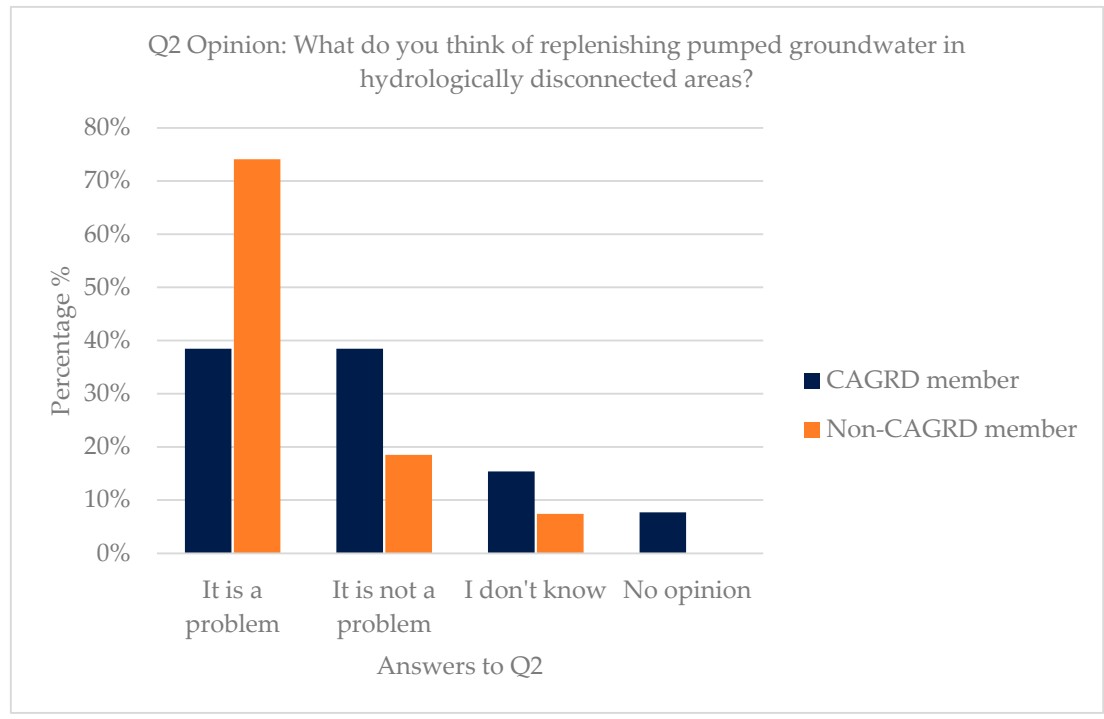

**Figure 6.** Distribution of answers given by CAGRD members and non-members to opinion question 2.

The hydrologic disconnect caused by recharge and recovery was ranked second, while the hydrologic disconnect caused by replenishment was ranked third. Finding alternatives to address the hydrologic disconnect caused by replenishment will be considered in drafting alternatives. Addressing issues of the hydrologic disconnect created by recharge and recovery will not be a priority because it is outside of the scope of the study.

3.2.3. Fourth-Ranked Issue: The AWS Program Allowing some Residential Developments and Municipalities in Central Arizona to Rely Primarily on Groundwater

Slightly more than half (55%) of respondents believed that the fact that the AWS program allows some residential development and municipal growth in Central Arizona AMAs to rely primarily on groundwater was a problem (Figure 7).

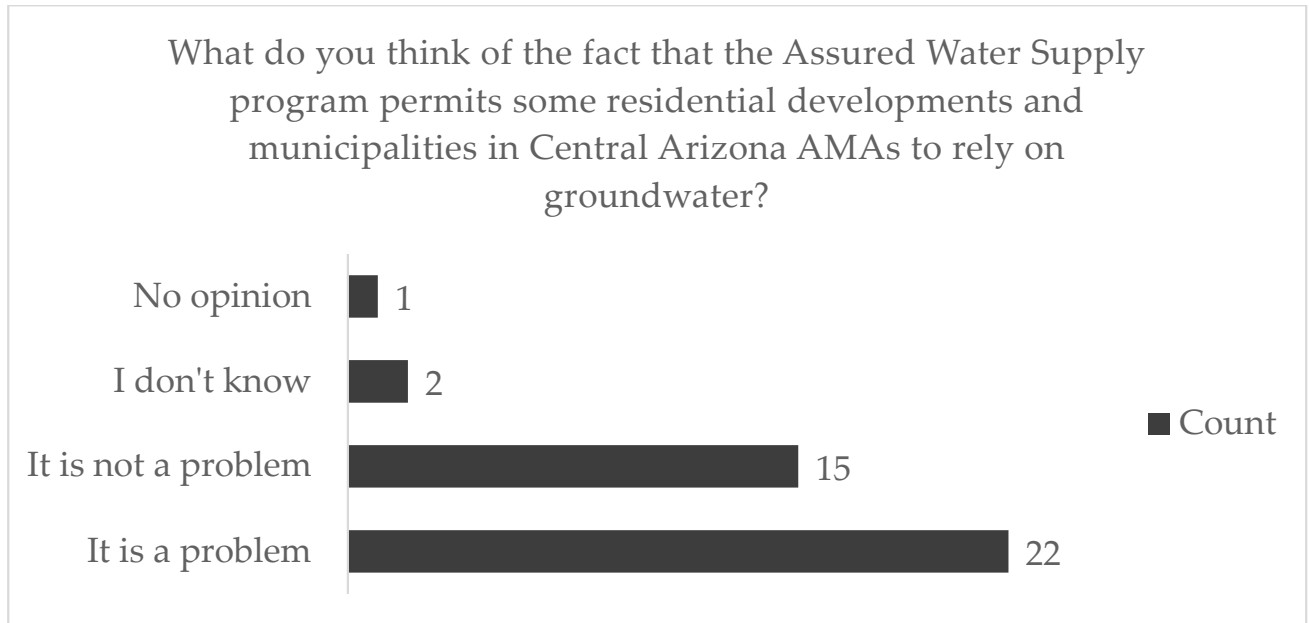

**Figure 7.** Survey answers to opinion question 3.

The results showed a relationship between the respondents' sectors and their opinions. The large majority of private entities (80%) believed that it was not a problem, while 67% of public entities and 75% of nonprofits believed that it was (Figure 8). However, it is important to note that the private sector represented only 26% of the stakeholder sample. If more entities in the private sector were to have answered the question, it is possible that the overall stakeholders' results for this issue would have been different.

This was the question that received the most comments. The comments showed that there was a large divide between opinions, with little to no middle ground. For some, using groundwater is an unsustainable practice because it allows pumping at unsustainable depths, and only a finite amount of groundwater can and should be used. For others, groundwater is a reliable supply and was characterized as a "much more drought resilient source of water than surface water" because it is not subject to fluctuating availability in the way that surface water supplies are. Additionally, one individual said that combining both sources of supplies is essential. Given that there was such a difference in opinions and because many argued that this problem goes beyond the CAGRD, reforms should address scales larger than that of the CAGRD. Therefore, addressing this issue does not fall within the scope of this work.

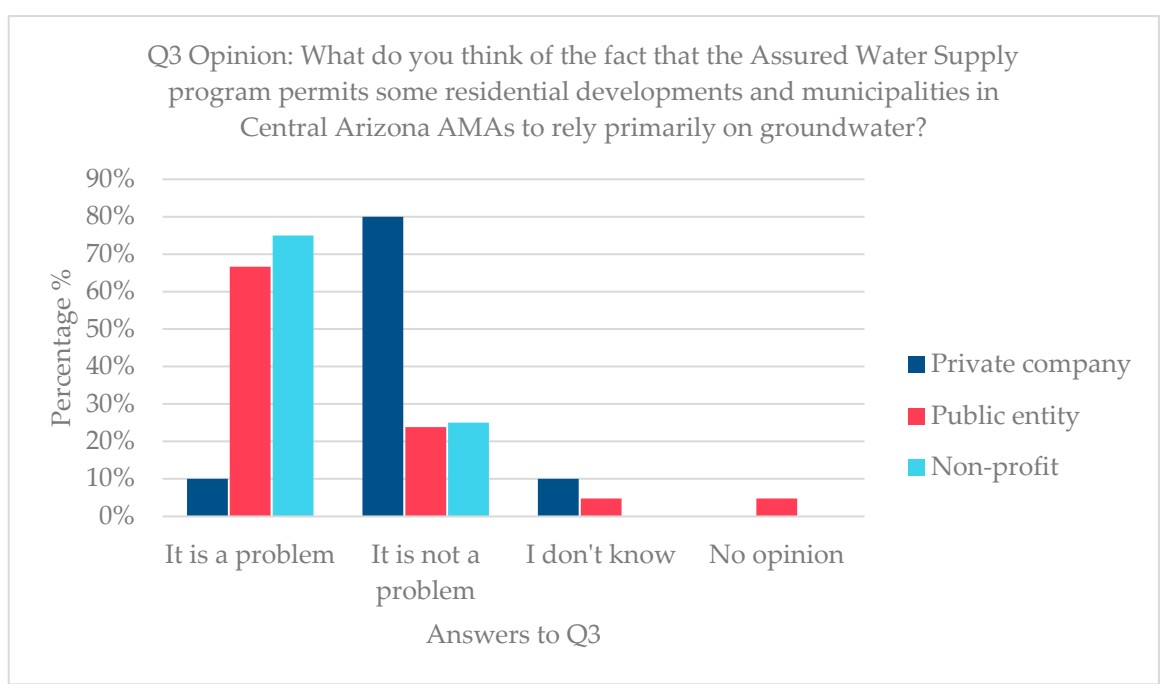

**Figure 8.** Distribution of answers given by CAGRD members and non-members to opinion question 3.

### 3.2.4. Fifth-Ranked Issue: CAGRD's Method of Charging Member Land Homeowners for Replenishment Services

The responses to the question about the current method of charging ML for replenishment services did not provide a conclusive assessment because 47.5% of respondents said that this was a problem, while 40% said that it was not (Figure 9). The results showed a relationship between the respondents' sectors and their opinions. In fact, this was the question that showed the most differences in opinion between stakeholders from the private, public, and nonprofit sectors (Figure 10). This question asked about issues that directly concern MLs. If we look at the survey responses from the ML category alone, two respondents said it was a problem, one said it was not, and one did not have an opinion.

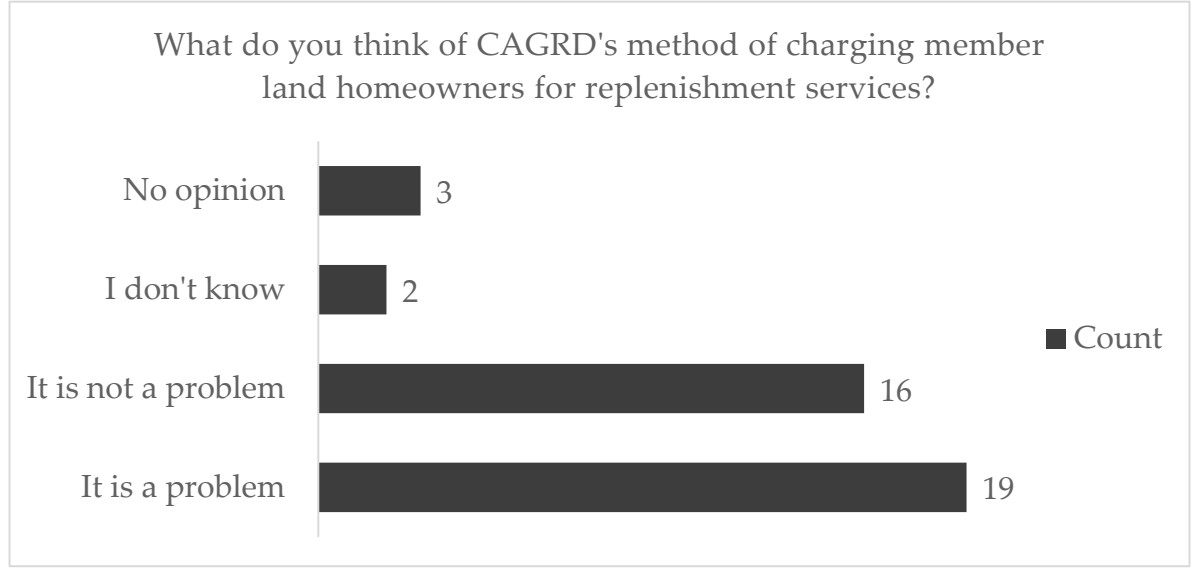

**Figure 9.** Survey answers to opinion question 5.

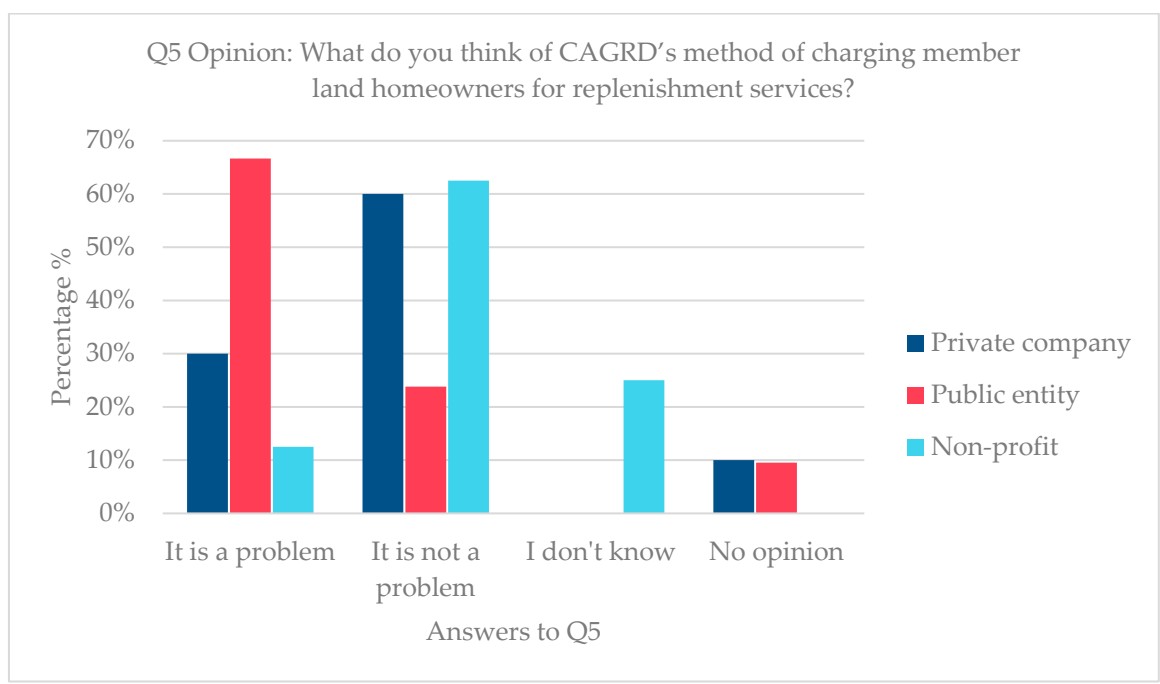

**Figure 10.** Distribution of answers given by CAGRD members and non-members to opinion question 5.

Based on the nature of the comments, responses to this question provide interesting insights into the logic of the stakeholders. For example, the comments did not respond to the method, but rather to the amounts charged, that is, the cost of water. Many stakeholders believed that developers should have to pay the upfront costs of providing a renewable water supply instead of passing those costs on to the CAGRD. Others were not concerned about the rising cost of water because they claimed that providing renewable water should not and cannot be cheap in the desert. As five individuals did not know or did not have an opinion on the issue, and because of the likelihood that stakeholders misinterpreted the question, these results were inconclusive; therefore, addressing this issue will not be the focus of this study.

3.2.5. Sixth-Ranked Issue: The Statutory Framework for ADWR and Review of the CAGRD Plan of Operation

About 50% of individuals believed that the statutory framework for the ADWR and review of the CAGRD Plan of Operation was not a problem, 15% believed it was a problem, and 35% did not know or chose not to give their opinion (Figure 11). Since such a large percentage of individuals did not give their opinion, we did not draw a conclusion from the survey responses but assumed that stakeholders generally did not consider ADWR review a problem.

The current statutory framework for ADWR review of the CAGRD Plan of Operation was the only matter which the majority of stakeholders did not believe to be a problem. A substantial percentage (35%) of respondents did not express an opinion. Three out of the six respondents who thought that this was a problem commented that they felt the ADWR would always approve the Plan for political reasons. More specifically, they said that the ADWR's opinion was not objective because it had political implications and influences. One insisted that an independent entity should review the Plan of Operation, and another said that the "ADWR should be free to give a technical opinion of the plan to flesh out the issues in future years and not worry about what it means for industry and economy, etc.". As there was no statistical difference between the opinions across stakeholders, the problem identified in this question was felt across stakeholder groups.

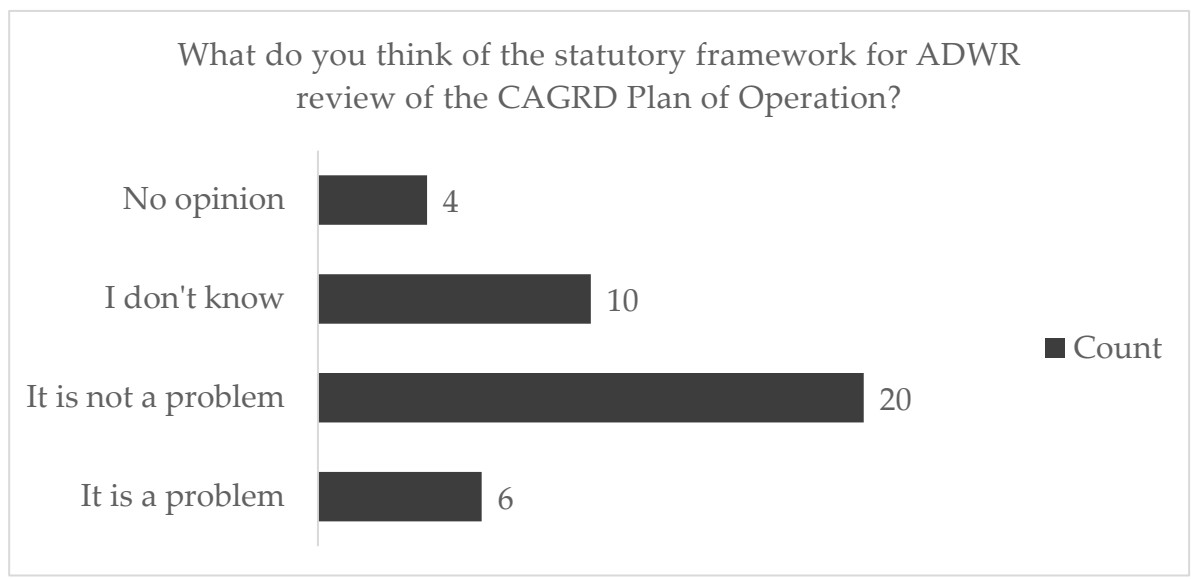

**Figure 11.** Survey answers to opinion question 6.

*3.3. Thematic Analysis of Experts' Solutions to CAGRD Problems*

In this section, we present seven proposed solutions discussed during the semi-structured interviews with experts about CAGRD issues and explain how relevant these solutions are to solving the problems highlighted by stakeholders in the survey. The experts were asked to provide solutions to what they believed the problems were (more experts might have agreed with these solutions but might not have said so or thought of these solutions at the time). This is why the solutions collected in Table 2 below do not necessarily address the issues deemed problematic by the majority of stakeholders. In Table 2, each column anonymously represents the answers of one expert. Columns also show when a particular expert provided a solution to more than one problem.

**Table 2.** Frequency of solutions mentioned by experts. Each column anonymously represents the answers of one stakeholder.

| Solutions to CAGRD Problems | Expert | | | | | | | | Total |
|---|---|---|---|---|---|---|---|---|---|
| Adjusting CAGRD Costs | ● | ● | | | | | | | 2 |
| Effluent Development | | | ● | | | ● | | | 2 |
| Infrastructure | | | | | | ● | | ● | 2 |
| Growth Culture | | | | ● | | ● | ● | | 3 |
| AWS Reform | | | | | ● | | | ● | 2 |
| Sub-AMAs | | | | | ● | ● | | | 2 |
| Transitional (Alternative) Entity | | | ● | | | ● | | ● | 3 |

### 3.3.1. Adjusting CAGRD Costs

One expert was concerned that the CAGRD would not have the financial means to pay for water acquisition in the future. This expert suggested raising the cost of CAGRD enrollment and the rates paid by homeowners. The expert explained that such a measure would have the benefit of discouraging some housing developers from enrolling their subdivisions with the CAGRD. This expert criticized developers, saying "they [developers] own the homes, sell them, and walk away, and they are gone, leaving behind an unknown financial obligation on the people who own the land in the district". Another expert explained that limiting membership would require the CAGRD to admit that it cannot take responsibility for more land development in the future.

From the standpoint of increasing the long-term viability of the CAGRD to provide water to citizens, limiting CAGRD membership would place less stress on the uncertainties related to the availability of renewable water supplies that are meant to meet the CAGRD's replenishment obligation. In addition, fewer replenishment obligations generate fewer hydrologic disconnects. Nevertheless, we believe this suggestion is not a solution that can be made actionable. Raising fees would require legislative action to change the foundation of the CAGRD's fee structure: the CAGRD cannot charge fees at levels that generate revenue in excess of its costs, that is, the CAGRD cannot make profits. Furthermore, the CAGRD has been accumulating water and saving it for future replenishment. In 2003, the Arizona Legislature created a replenishment reserve in each of the Phoenix, Tucson, and Pinal AMAs to allow the CAGRD to purchase and accumulate long-term storage credits in anticipation of rising water rates [31].

### 3.3.2. Effluent Development

Experts suggested that the CAGRD should focus on increasing water supply by reusing more effluent. As one interviewee stated, such an approach would "expand the amount of water available to live in the desert". Across the interviews conducted, experts suggested three different ways for the CAGRD to reuse effluent. First, more wastewater treatment plants should deliver effluent to golf courses and common green areas of master-planned communities. Second, new effluent recharge projects can be developed through partnerships. As one expert said, "one example [of a partnership] is in the West Valley with Liberty Utilities and CAP. They partnered on an effluent recharge project. I think that is innovative, I think that is the way things have to go. It takes the effluent that is being generated by CAGRD members and replenishes it basically in the area of impact". Indeed, the effluent generated at Liberty's Palm Valley Water Reclamation Facility is recharged in USF basins [69]. Third, the state of Arizona might invest in cost-effective technology for direct potable reuse. The rationale is that recharging activities might become too cost-prohibitive.

Certainly, bringing a new supply would address the uncertainties related to the availability of renewable water supplies to meet the CAGRD's replenishment obligation. Encouraging more effluent reuse for outdoor areas is prudent. Although the area of green lawns in Phoenix decreased by 82% between 2000 and 2019 [70], homeowners in Central Arizona use up to 70% of their domestic water outside their houses mainly on landscape irrigation [71]. Using less potable water for landscaping and recharging effluent could reduce the hydrologic disconnect and help residential developments and municipalities rely on less groundwater, but only if some infrastructures are developed to transport effluent to green spaces in residential areas. This is why the suggestion of effluent development will be used to draft alternatives in Section 4.

While developing direct potable reuse would solve some of these same issues, it is speculation that recharge activities might become more expensive than the treatment of effluent to drinking-water standards, especially considering the price of infrastructure to transport that water.

### 3.3.3. Infrastructure

The CAGRD was created to allow subdivisions to be built in areas lacking renewable water resources. According to one expert, when the CAGRD was created, subdivisions were expected to ensure direct renewable water delivery in the long term and reduce reliance on the CAGRD. However, as several interviewees stated, this expectation remains unmet: "the infrastructure to bring renewable supplies to these subdivisions has not been created so they are relying on the replenishment of the groundwater for the groundwater use to be consistent with the management goal and so they don't have the ability to receive renewable supplies directly". The creation of long-term renewable supplies has not materialized, as this would have required constructing expensive infrastructure from the CAP canal and potentially extending or modifying the CAP. As one expert said: "The issue is that it is

easier to develop the way it is. It is cheaper, it is easier and it has really been good to the development community and the homebuilders, that is the big issue".

For example, subdivisions might have funded the construction of conveyance infrastructures leading from the CAP into their local supply networks. The experts interviewed in this study were adamant that piping infrastructure is the solution to limit subdivisions' reliance on groundwater despite the costs and that it must be achieved in the near future, now that the CAGRD issues have gone on for several decades.

Experts added that infrastructure costs result from land-use decisions. As one interviewee explained, "in Arizona, land is not always cheap where there is water available. And development communities want to develop where they can make the most money". Real-estate developers therefore build where costs are low relative to the demand for their houses—often on undeveloped desert distant from access to renewable supplies.

Building recharge facilities or being able to transport renewable water to where it is needed would reduce the hydrologic disconnect and decrease reliance on groundwater by some residential development. Since building additional infrastructure is an essential complement to CAGRD-specific solutions, it will be considered in drafting alternatives in Section 4.

### 3.3.4. Growth Culture

Three CAGRD experts questioned the statewide policy of growth in Arizona. For example, one expert raised the following question: "Should we support growth at all cost, or do we stop growing? [ . . . ] what is reasonable growth?". For another expert, reasonable growth is the opposite of the sprawl machine. In this context, CAGRD is seen as enabling unsustainable practice: "[CAGRD] drives unsustainable development and I think it has helped to facilitate faster development of sprawl style development on undeveloped desert instead of more compact development in communities, so it is part of the sprawl machine". Boyer et al. [72] explain that authors such as Logan [73] and Gober et al. [74] have suggested policies of land-use regulation and the limitation of urban sprawl since 1970. Robbins [70] uses the example of the metropolitan area of Phoenix to describe an "unrelenting sprawl". The same expert who commented on the sprawl machine added that "growing fast does not mean that we are growing in a way that is sustaining people, creating a good quality of life [ . . . ]. Mechanisms like the CAGRD feed into that because again, they are fairly short-sighted".

The comment of another explains the appearance of short-sightedness as being due to the fact that the CAGRD was supposed to be a temporary solution: "My perspective on this is CAGRD was put in place as an incremental solution to a longer-term problem [ . . . ] CAGRD is an incremental solution that I don't know was really meant to be the ultimate for every solution [ . . . ] I hope, I don't think anybody ever thought CAGRD was the solution to all the problems. It was one step toward the long-term solution".

Two experts pointed out that growth is a global state issue beyond the CAGRD and that land-use policy seems to be an appropriate way of approaching water sustainability problems. Indeed, one interviewee said: "[Growth] is not really a CAGRD issue, that is a state issue", and that the "ADWR is going to have to get tough about where there is not enough physically available water to allow continued growth". Another expert explained, "there is a land-use policy that frankly seems a better way to approach those issues than going after the CAGRD". In the literature, Molotch believes that a creative land-use program could emerge if government would ask "what it can do for its people rather than what it can do to attract more people" [75]. In other words, Arizona's policymakers would have to shift their priorities from coinciding with the development community that wants to bring new people to the state to pleasing Arizonans. Molotch also criticizes growth and the fact that increased costs "caused by new development [must] be borne by the public at large, rather than by those responsible for the "excess" demand on the urban infrastructure" [75]. This raises the question of whether real-estate developers and home builders should be the ones paying to build infrastructure.

Limiting growth would limit the CAGRD's future replenishment obligation, the hydrologic disconnect, as well as some of the reliance by residential development and municipalities on groundwater. However, changing policies supporting the culture of growth in Arizona is clearly a solution outside the CAGRD-specific scope of this research. While CAGRD policy changes that affect the AWS approval of development can have an impact on growth, countervailing forces remain strong. This is why this suggestion will not be selected to draft policy alternatives.

### 3.3.5. Assured Water Supply Reform

Two experts suggested amending the AWS Rules. One expert said that rather than "a change to the CAGRD, it is more a change to the AWS Rule" that would be necessary to "provide some additional incentive for undesignated water provider serving ML to become designated or try to acquire other renewable supplies". The implication is that these water providers would deliver renewable water instead of groundwater to subdivisions. However, many designated water providers rely on the CAGRD, so this policy suggestion would not necessarily address any of the CAGRD issues. The second suggestion would be to modify the AWS Rules to accommodate new AWS criteria. Another expert had this to say: "part of what drives people to the CAGRD, and the AWS Rules are not too strict necessarily, but the kind of supply that would qualify for an AWS are mostly already taken and gone [ . . . ] whatever we come up with then, the AWS Rules would have to be modified to accommodate what we have decided [is] a good enough supply for new growth".

Interview analyses did not reveal a consensus on what the experts meant by amending the AWS Rules. If the new criteria consisted of implementing less stringent regulations by allowing more groundwater pumping, this would limit reliance on the CAGRD and therefore decrease the CAGRD's future replenishment obligation. However, such a proposal would exacerbate other stakeholder-identified problems associated with the CAGRD, such as localized groundwater depletion.

### 3.3.6. Sub-Active Management Area Sub-Basins

Two experts considered creating sub-AMAs, that is, administrative divisions within the already existing AMAs: "You cannot manage water, groundwater over an entire AMA, you have to manage it over smaller portions, smaller hydrologic areas". Moving from AMA-wide goals to AMA sub-basin goals would limit the hydrologic disconnect between basins. In essence, this strategy would allow water managers to concentrate their attention on particularly problematic areas. While this solution could have effects beyond the CAGRD-specific problems, it could reduce the hydrologic disconnect of replenishment, especially if it influences infrastructure development and land-use decisions.

### 3.3.7. (Transitional) Alternative Entity

According to the experts, the CAGRD has the monopoly of helping entities that do not have enough renewable supplies to gain AWS approval. Three experts suggested creating another entity that could move development away from the CAGRD. One expert characterized the CAGRD as a transitional entity: "The only way those cities [like Buckeye or Queen Creek] can grow their tax base to generate the revenue and go buy their water supplies is through the CAGRD, the CAGRD is a stepping stone".

An alternative entity could be a collective entity (a cooperative arrangement, joint-action district, public agency, or private entity) intended to acquire and organize water supplies and share them among its members to meet the AWS requirements. Such an entity would increase the capacity of municipal water providers to negotiate to obtain legal rights regarding water acquisition. Water users who are too far from existing infrastructures could temporarily turn to the CAGRD, giving cities time to build infrastructures to later receive renewable water directly through the alternative agency, or the CAGRD could be a stepping stone for some municipal water providers to help them grow and collect more revenue to build infrastructures later, deliver renewable water, and acquire their

designation of AWS. Several cities relied on the CAGRD for a regulatory purpose while their goal was ultimately to use renewable supplies. Water users could then cancel their CAGRD membership. The proposed entity could also facilitate wheeling agreements to transport non-project water through the CAP aqueduct. In the end, the CAGRD would become a transitional mechanism rather than a permanent water-supply solution.

The experts disagreed, however, on how this entity would be governed. One expressed that they would like to see water markets emerge: "My sense though is that [an] institution might emerge, more about private markets, there are folks out there with deep pockets looking to make money on water resources. And there is a lot of money to be made on development". Another said that they would not want that kind of market: "What I also would not want to see is the privatization of water rights, that you create speculative interest that started buying all the water rights and they were kind of held by a private sector, I think that's just asking for trouble [ . . . ] But having a private entity in there to me is ok as long as they don't control or dominate the market".

Although this suggestion could reduce reliance on the CAGRD for replenishment, it would not reduce uncertainty about the availability of supplies. Some efficiencies may be achieved that stretch existing supplies. However, the devil is in the details, which remain vague. The introduction of a water market or another new institution does not depend on changes to CAGRD policy and is therefore beyond the scope of this research.

*3.4. Error Analysis and Limitations*

To limit errors of non-observation associated with the sampling method, the persons participating in a survey must have characteristics similar to those of the relevant population. The stakeholder selection may have been biased because we could not find an email for every stakeholder identified in the stakeholder list. Sampling numbers differed between different stakeholder groups because of logistical issues associated with finding contact information, as well as being based on the population demographics of the different groups. Although a great effort was made to call different entities and request an email address to send the survey to, we sampled stakeholders it was convenient to contact. Furthermore, we sent follow-up emails twice to encourage non-respondents.

Since this survey measured subjective mental states, such as attitudes, opinions, and feelings among stakeholders, rather than objective facts, there is an element of measurement error associated with this study. While these errors cannot be measured, we tried to limit them when designing the questions. For example, to cover cases in which individuals might not have an answer or not feel comfortable sharing their views, we offered the answers "I don't know" and "No opinion", as well as the option of skipping opinion questions. Moreover, errors of distortion should also be taken into account. Some stakeholders might have decided to provide answers that aligned with their agendas which did not necessarily reflect their genuine opinions.

This study experienced missing data at random. Although the opinion questions were optional, forty respondents answered them. Only one respondent left all of the questions blank (but submitted comments). Missing data were not consistent with respect to professional category, sector, or membership status. As this study did not experience missing data "not at random", we found support in the data obtained when respondents did not skip questions strategically, and thus the results did not overstate certain opinions.

Finally, the random sample of each category of stakeholders was representative of the population surveyed, with the exception of Native Nations. Indeed, Native Nations are under-represented in the results of this study because none of the seven Native Nations surveyed chose to participate. Furthermore, the sample size was the same for each opinion question, which means that the importance of issues can be compared at the same level.

## 4. Discussion of Policy Alternatives to Address CAGRD Issues

This section offers two solutions based upon the solutions suggested by the experts described above: effluent and infrastructure development and limiting CAGRD enrollment

to designated AMA sub-basins. These two solutions aim to address two CAGRD specific problems that are top priorities for stakeholders: the uncertainties of CAGRD supplies and the hydrologic disconnect caused by replenishment activities. The first and second policy alternatives would partially address the problem of the uncertainties of CAGRD supplies by limiting demand for replenishment. The second policy alternative would address some of the problems related to the hydrologic disconnect.

### 4.1. Policy Alternative 1: Using Effluent for Replenishment

The objective of this policy alternative is to help with the uncertainties of CAGRD supplies by connecting CAGRD with renewable water. It could also reduce CAGRD replenishment obligations and contribute to mitigating the effects of the hydrologic disconnect between groundwater pumping by CAGRD members and replenishment.

According to Avery et al. [10], when designing their subdivision, developers must plan how to take back part of the effluent generated by a local wastewater treatment plant. This incentivizes developers to build green areas in their subdivision to dispose of this effluent. We suggest that wastewater treatment plants partner with the CAGRD to recharge the effluent generated by a subdivision. The CAGRD would partner to develop new recharge facilities near wastewater treatment plants and purchase effluent from plants for replenishment. This would increase the supply of water available to the CAGRD. Assuming that wastewater treatment plants are near the areas where groundwater is pumped, effluent recharge would mitigate the hydrologic disconnect. In addition, developers would be able to reduce the area of grass designed for effluent disposal, which would reduce total water use by the development.

This policy alternative would require the CAGRD to obtain a recharge facility permit and water-storage permit from the ADWR and an Aquifer Protection Permit from the Arizona Department of Environmental Quality for each replenishment facility created in this way. The CAGRD would receive credits for recharge. The Central Arizona Water Conservation District board would have to decide on a reduced fee for developers [76].

It is important to note that the question of effluent quality might need to be addressed before considering this solution. While the effluent used for aquifer recharge must be of the highest quality [77,78] in Arizona, many compounds are unregulated by federal and state governments and may be found in groundwater.

### 4.2. Policy Alternative 2: Limit Membership Enrollment to Selected Sub-Basins

The main objective of this policy alternative is to mitigate the hydrologic disconnect between the area in which groundwater is pumped by or for CAGRD members and the area in which groundwater is replenished by the CAGRD. This policy alternative may also temporarily reduce future CAGRD obligations, as development adjusts to the new rules, therefore helping with water supply uncertainties. This policy would delineate sub-basins in each AMA and limit new CAGRD enrollment to developments located in sub-basins in which the CAGRD has the means to replenish water. According to the ADWR (2021), there are seven sub-basins in the Phoenix AMA, five sub-basins in the Pinal AMA, and two sub-basins in the Tucson AMA (Figure 12).

This policy alternative would require replacing the "active management area" phrasing with "sub-basin of an active management area" in eleven sections of the Arizona Revised Statutes [33,34,79–88]. The statute [34] requiring groundwater withdrawal in the east or west portion of the AMA to be replenished in that same portion would become unnecessary and would be deleted. Moreover, this policy alternative would require the CAGRD to amend its membership enrollment statutes by limiting enrollment to members located in a sub-basin of an AMA in which the CAGRD has methods of replenishment [89].

In practice, the CAGRD would need to develop new recharge facilities or seek partnerships that would facilitate replenishment in sub-basins where storage capacity is not available. In the Phoenix AMA, the CAGRD does not have facilities in the Fountain Hills sub-basin, where CAGRD members are located. The CAGRD does not have facilities in

the Lake Pleasant and Carefree sub-basins, but it does not have CAGRD members there. In the Pinal AMA, the CAGRD has replenishment obligations for members located in the Maricopa Stanfield and the Eloy sub-basins. The CAGRD has identified two large facilities to replenish groundwater in these two sub-basins. However, this policy could limit membership in the three other Pinal AMA sub-basins because there are no facilities there. In Tucson AMA, the CAGRD owns a re-charge facility in each of the two sub-basins.

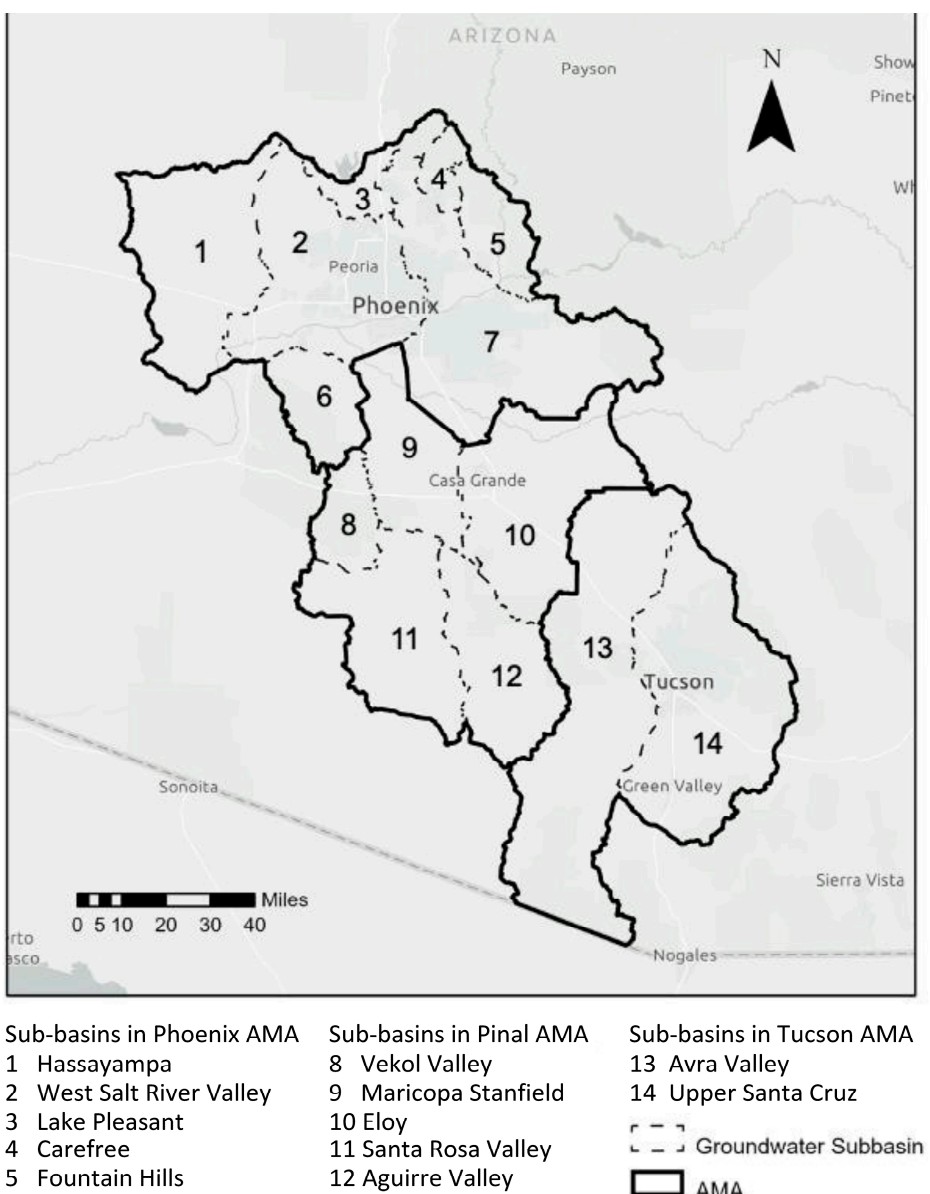

**Sub-basins in Phoenix AMA**
1 Hassayampa
2 West Salt River Valley
3 Lake Pleasant
4 Carefree
5 Fountain Hills
6 Rainbow Valley
7 East Salt River Valley

**Sub-basins in Pinal AMA**
8 Vekol Valley
9 Maricopa Stanfield
10 Eloy
11 Santa Rosa Valley
12 Aguirre Valley

**Sub-basins in Tucson AMA**
13 Avra Valley
14 Upper Santa Cruz

└ ─ ┘ Groundwater Subbasin

▭ AMA

**Figure 12.** Map with sub-basins delineated in the Phoenix, Pinal, and Tucson AMAs.

## 5. Conclusions

The CAGRD has been an essential mechanism enabling new developments and municipal water providers to show consistency with the management goals of the Phoenix, Tucson, and Pinal AMAs and obtain an AWS. This article reports the results of a survey of 41 Arizona stakeholders designed to elicit their opinions on six CAGRD-related issues and offers policy alternatives derived from sixteen interviews that address the CAGRD issues. The two policy alternatives proposed would also contribute to the solution of broader issues. While both policy alternatives would involve practical and legal challenges, they

have some support among experts on the CAGRD. However, this study shows that a lack of consensus among stakeholders risks hindering support for any specific policy.

The lack of consensus across stakeholder sectors hinders solution development. On two issues—long-term supply uncertainties and continued growth based on groundwater—the public and private sectors appear to disagree. Public- and private-sector entities have different constituencies and financial circumstances, and consequently they have different motivations and perspectives. Beyond CAGRD-related questions, stakeholders appear to be concerned about groundwater overdraft, water shortage and scarcity, and unsustainable groundwater pumping practices. The CAGRD has a relatively restricted role in these larger issues.

The results of this work suggest several ideas and perspectives for future studies. A larger stakeholder survey seeking to characterize respondents who hold differing opinions could provide useful information for policy discussions. Nevertheless, the results of the expert interviews highlight the underlying question of land-use decision making, which was never raised as an issue by the stakeholders. Approaching water issues from the perspective of land use would provide fresh insights toward solving water challenges in Arizona. Likewise, a prospective opinion study should be conducted to evaluate the political feasibility of policy alternatives. Such future research would engage CAGRD experts and stakeholders in informing the Arizona policy-making process. Furthermore, additional research should be conducted to examine how effluent recharge may impact groundwater quality. In particular, the likelihood of finding contaminants of emerging concern, such as pharmaceuticals, in groundwater used for potable use should be investigated.

This article informs current and future discussions about groundwater management in Central Arizona aimed at securing a sustainable water supply. While the findings of this article are case-specific, the innovative methodology developed could be easily transferred to evaluate existing policies with respect to emerging problems worldwide.

**Author Contributions:** Conceptualization, R.F.A.B., S.B.M. and S.E.; methodology, R.F.A.B., S.B.M. and L.A.B.; software, R.F.A.B.; validation, R.F.A.B.; formal analysis, R.F.A.B.; investigation, R.F.A.B.; resources, R.F.A.B., S.B.M. and L.A.B.; data curation, R.F.A.B.; writing—original draft preparation, R.F.A.B.; writing—review and editing, R.F.A.B., S.B.M., S.E. and L.A.B.; visualization, R.F.A.B.; supervision, S.B.M.; project administration, R.F.A.B., S.B.M. and S.E. All authors have read and agreed to the published version of the manuscript.

**Funding:** This research was supported in part by the Technology Research Initiative Fund administered by the University of Arizona Office for Research, Innovation, and Impact, funded under Proposition 301, the Arizona Sales Tax for Education Act in 2000.

**Institutional Review Board Statement:** This project has been reviewed and approved by an IRB Chair or designee (protocol number: 2008926930).

**Informed Consent Statement:** Informed consent was obtained from all subjects involved in the study.

**Data Availability Statement:** Data is unavailable due to privacy and ethical restrictions.

**Acknowledgments:** The authors would like to thank Evy Lizzaraga, from Information Technology Support at the University of Arizona, for helping organize the data results of the Qualtrics survey. The authors would also like to thank Rebecca Blakiston, at the University of Arizona Libraries, for helping design the survey.

**Conflicts of Interest:** The authors declare no conflict of interest. Rebecca Bernat is now the Technical Administrator for the Arizona Water Banking Authority and an employee of the Arizona Department of Water Resources. The supporting research was completed prior to her current employment. This article does not represent the views of the Arizona Department of Water Resources or the State of Arizona.

## Abbreviations

AMA      Active Management Area
AWS      Assured Water Supply
CAGRD   Central Arizona Groundwater Replenishment District
ML        Member Land
MSA      Member Service Area
CAP      Central Arizona Project

## Appendix A

**Table A1.** Survey questions to collect information about the dependent variables.

| Question | Mutually Exclusive Answers (Optional) | Option to Comment |
|---|---|---|
| Question 1: What do you think of recharging water (at underground storage facilities or groundwater savings facilities) and recovering it in hydrologically disconnected areas in Central Arizona AMAs? | It is a problem<br>It is not a problem<br>I don't know<br>No opinion | Yes |
| Question 2: What do you think of replenishing pumped groundwater in hydrologically disconnected areas? | It is a problem<br>It is not a problem<br>I don't know<br>No opinion | Yes |
| Question 3: What do you think of the fact that the Assured Water Supply program permits some residential developments and municipalities in Central Arizona AMAs to rely primarily on groundwater? | It is a problem<br>It is not a problem<br>I don't know<br>No opinion | Yes |
| Question 4: What do you think of the long-term uncertainties related to the availability of renewable water supplies to meet CAGRD's replenishment obligation? | It is a problem<br>It is not a problem<br>I don't know<br>No opinion | Yes |
| Question 5: What do you think of CAGRD's method of charging member land homeowners for replenishment services? | It is a problem<br>It is not a problem<br>I don't know<br>No opinion | Yes |
| Question 6: What do you think of the statutory framework for ADWR review of the CAGRD Plan of Operation? | It is a problem<br>It is not a problem<br>I don't know<br>No opinion | Yes |

**Table A2.** Final survey question to rank CAGRD issues by priority order.

| Question | Optional Responses, Not All Statements Have to Be Ranked | Option to Comment |
|---|---|---|
| Rank the following issues by order of priority (1 being highest priority, and 6 being lowest priority): | Recharging water and recovering it in hydrologically disconnected areas in Central Arizona AMAs | No |
| | Replenishing pumped groundwater in hydrologically disconnected areas | |
| | The Assured Water Supply program permitting some residential developments in Central Arizona AMAs to rely primarily on groundwater | |
| | Long-term uncertainties related to the availability of renewable water supplies to meet CAGRD's replenishment obligation | |
| | CAGRD's method of charging member land homeowners for replenishment services | |
| | The statutory framework for ADWR review of the CAGRD Plan of Operation | |

## Appendix B

**Table A3.** Survey questions to collect information about the independent variables.

| Question | Mutually Exclusive Answers (Mandatory) | Option to Comment |
|---|---|---|
| What category do you primarily represent or identify with? | Academic | No |
| | Consultant | |
| | Government entity | |
| | Homebuilder | |
| | Industry | |
| | Investment firm | |
| | Municipal water provider | |
| | Native Nation | |
| | Non-governmental organization | |
| | Real estate developer | |
| | Other (please specify) | |
| Do you want to select another category? | Yes | No |
| | No | |
| If yes, what other category do you represent or identify with? | Academic | No |
| | Consultant | |
| | Government entity | |
| | Homebuilder | |
| | Industry | |
| | Investment firm | |
| | Municipal water provider Native Nation | |
| | Non-governmental organization | |
| | Real estate developer | |
| | Other (please specify) | |

**Table A3.** *Cont.*

| Question | Mutually Exclusive Answers (Mandatory) | Option to Comment |
|---|---|---|
| What sector do you represent or identify with? | Private company | No |
| | Public entity | |
| | Non-profit | |
| | Other (please specify) | |
| Are you (or do you represent) a CAGRD member? | Yes, member service area | No |
| | Yes, member land | |
| | No | |
| Please select your County: | Maricopa | No |
| | Pima | |
| | Pinal | |
| | Other (please specify) | |

## Appendix C

**Table A4.** List of questions asked to CAGRD experts during interviews.

| Questions for Semi-Structured Interviews with CAGRD Experts |
|---|
| 1. What is your current position and what are your primary responsibilities? |
| 2. What experience have you had with CAGRD in your current or former positions? |
| 3. In your experience, what have been the pros and cons of the way CAGRD operates? |
| 4. Based on a literature survey, I have identified two main issues to CAGRD: (1) unlimited growth in membership leading to increasing groundwater replenishment obligations, and (2) the hydrologic disconnect between water pumping by CAGRD members and replenishment. Are you familiar with these two issues, and do you think they sufficiently describe the issue of CAGRD? Are there additional issues that should be added to the list? If so, what are they? |
| 5. What policies could be implemented as solutions to each issue we talked about? Note: These policies do not have to target CAGRD only. |
| 6. What else would you like to share about CAGRD? |
| 7. I will survey stakeholders about the policies we have discussed to evaluate their support. Who (individual or organizational name) should be getting such survey? |
| 8. Who are CAGRD experts you would recommend I interview? May I share your name with these individuals when I contact them? |

## Appendix D

**Table A5.** Distribution of the survey sample vs. survey respondents.

| | Individuals Surveyed | | Respondents | |
|---|---|---|---|---|
| | Number | % | Number | % |
| Academic | 4 | 3% | 1 | 2% |
| Consultant | 7 | 5% | 6 | 15% |
| Governmental entity | 40 | 29% | 12 | 29% |
| Homebuilder | 2 | 1% | 0 | 0% |
| Industry | 6 | 4% | 2 | 5% |
| Investment firm | 2 | 1% | 0 | 0% |
| Municipal water provider | 37 | 27% | 11 | 27% |
| Native Nation | 7 | 5% | 0 | 0% |
| Non-governmental organization | 10 | 7% | 3 | 7% |
| Real-estate developer | 2 | 1% | 1 | 2% |
| Other | 20 | 15% | 5 | 12% |
| Total | 137 | 100% | 41 | 100% |

## Appendix E

**Table A6.** Presentation of the topics and arguments mentioned by stakeholders for each issue. Each column of Table A6 anonymously represents the answers of one stakeholder. Columns also show when a particular respondent commented on more than one issue.

| Comments | Respondents | | | | | | | | | | | | | | | | | | | | | | | Total |
|---|---|---|---|---|---|---|---|---|---|---|---|---|---|---|---|---|---|---|---|---|---|---|---|---|
| **Issue #1** | | | | | | | | | | | | | | | | | | | | | | | | |
| Groundwater overdraft | • | | • | • | | | | | | | • | | | | | • | | | | • | • | | | 7 |
| Water recovery | | | | | | | | | | | | | | | | | | | | • | • | | | 2 |
| Benefits | | | | | | | | | | | | | | • | | | | • | | | | | | 2 |
| Minor issue | | • | | | | | | | | | | | | • | | | | | | | | | | 2 |
| **Issue #2** | | | | | | | | | | | | | | | | | | | | | | | | |
| Long-term impact | | | | | | | • | | | • | | | | | | | | | | | | | | 2 |
| Benefits | | | | | | | • | | | | • | | | | | | | | | | | | | 2 |
| Minor issue | | | | | | | | | | • | | | | • | | | | | | | | | | 2 |
| **Issue #3** | | | | | | | | | | | | | | | | | | | | | | | | |
| Unsustainable | • | | | | • | | | | | | | | • | | | • | | | | | | | | 4 |
| Groundwater reliability | | | | | | | | | | • | | | | • | | | | | | | • | | | 3 |
| **Issue #4** | | | | | | | | | | | | | | | | | | | | | | | | |
| Shortages and scarcity | • | | | | | | | • | | | | • | • | | • | • | | | | | • | | | 7 |
| Supply competition | • | | | | | | | | | | | | | | | | | | • | | | | | 2 |
| Cost | • | | | | | | | | | | | | | | | | | | | | | • | | 2 |
| Broader issue | | | | | | | • | | • | | | | | | | | | | | | | | | 2 |
| Effluent reuse | | | | | | • | | | | | | | | | | | • | | | | | | | 2 |
| Desalination | | | | | | • | | | | | • | • | | | | | | | | | | • | | 4 |
| **Issue #5** | | | | | | | | | | | | | | | | | | | | | | | | |
| Developer cost | | | | | • | | • | | | | | | | | | • | | | | | | | | 3 |
| True cost of water | | | | | | • | | | | | | | | | | • | | | | | | | | 2 |
| **Issue #6** | | | | | | | | | | | | | | | | | | | | | | | | |
| Politics | | | | | | | | | | | | | • | | | • | | | | • | | | | 3 |

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
