# Peer review of "Stakeholder Opinions on the Issues of the Central Arizona Groundwater Replenishment District and Policy Alternatives"

_water, doi:10.3390/w15061166_

Round 1

Reviewer 1 Report

This paper is very interesting and is a major contribution to scientific management of groundwater and ensuring sustainable supply of water resources. It takes Arizona, which is at the forefront of groundwater management, as an example. And the statistical and inductive thematic content analysis method used in this paper has certain innovation. From the perspective of various stakeholders, six difficult problems need to be solved were identified. Finally, special analysis was made for these problems, providing theoretical reference for scientific management of groundwater in other regions. The structure and argument of this paper are clear. The paper is well-written and appropriate for the journal, but should be improved by addressing the following aspects.

1.The introduction provides a detailed description of the actual situation by introducing background, water supply regulations in Arizona, literature and policy gaps, and central Arizona groundwater replenishment district issues. However, the introduction part is too long, and there is a lack of logic and coherence between each section. It is suggested that the introduction part should be optimized.

2.There are two 2.3.2 in the paper. It is suggested to confirm and arrange the serial number.

3.As mentioned in 3.4, this survey is based on subjective ideas rather than objective facts, which has certain errors and does not necessarily reflect the true opinions of the interviewees, so whether other measures can be taken to avoid such errors and make the research more scientific.

4.The section 4 and 5 propose two solutions based on the solutions proposed by experts, providing policy recommendations for groundwater management in Arizona. However, the suggestions are only for the situation in Arizona. Can the author provide reliable theoretical reference for groundwater management in other countries or regions based on the case of Arizona?

5.The clarity of some pictures in the paper is not enough, so it is suggested to increase the DPI.

Author Response

Dear Reviewer 1,

We would like to sincerely thank you for your comments, and suggestions to improve this article. Please see an explanation of the changes we made on page 1 of the attached pdf.  Changes in the documents were made using track-changes.

Reviewer 2 Report

Given the nature of the information, this manuscript is way too long. In my opinion it needs to be reduced by 70% and focus on presentation of the opinions of the very small number of respondents.

To be published, there is also a need for a conclusion or set of findings that are conclusive in the sense that something other than do more research is needed.

At present the manuscript reads like a consultants report that presents the data in a form that is difficult to understand. It is for example, unrealistic to expect a reader to remember how question 1 was defined many pages earlier.

I think the information presented can be presented in around 10 pages.

Author Response

Dear Reviewer 2,

Thank you for your comments, and suggestions to improve this article. Please see our explanation on page 6 of the attached pdf.  Changes in the word document were made using track-changes.

Reviewer 3 Report

The article deals with an interesting form of groundwater management in Arizona. The introduction describes in detail (even a bit too detailed) how the management is organized there. The description is vivid and easy to follow, although one is burdened with an abundance of individual information and abbreviations (elf).

The article describes an interesting case study. It reads like a well-written report of a consulting project. The research questions presented in the introduction are important questions for practical groundwater management in the region. However, I wouldn’t consider the paper as a scientific paper suitable for a scientific journal. The stated research questions are not scientific questions for the following reasons:

- They are not designed to advance the state of the science. In fact, the article does not systematically review the literature and the state of the art of groundwater management research in other regions and countries and relate the their work to the literature.

- It is not elaborated what innovations the article brings.

- Only questions related to the specific case study are asked. There is no reflection on whether the answers to the questions are transferable to other cases.

The introduction (lines 189-195) further introduces a distinction between issue (a topic that needs debating) and problem (a problem requires solving). It is claimed that this distinction is important in order to conduct policy analysis. This distinction is not separative, but confusing. In fact, the terms are sometimes used interchangeably in the rest of the text (e.g., line 318, line 385, lines 905-914).

The two policy options dubbed in the Discussion and Conclusion as key proposals and findings of the paper are already the result of a literature review (Sec. 1.3, see also Appendix C, Question 4) prior to the start of the survey. In this respect, they were merely confirmed by the survey. 

Some further, specific comments are annotated in the manuscript.

Author Response

Dear Reviewer 3,

We would like to sincerely thank you for your comments, and suggestions to improve this article. Please see an explanation of the changes we made on pages 2-6 of the attached pdf.  Changes in the word document were made using track-changes.

Round 2

Reviewer 2 Report

I would like to see the word "should" replaced with "could" at line 1036 and 1038.

I am don't believe that the authors are in a position to preach given the qualifications around their research.

I think that it is just worthwhile letting this paper go through.  I would have liked to have seen more radical shortening and more consideration of implications beyond the resource studied

Reviewer 3 Report

My main critique of the paper is that it lacks an appropriate literature review. Section 1.3 does not systematically review the literature on the role of stakeholders in water resources management. The cited literature refers almost exclusively on the specific case in Arizona. The remaining quotes describe only very general relationships (citation 10, 11) and definitions (citation 12-15, 33, 34). This is insufficient. One would expect At this point, one would expect a classification of the case study and its methodology in the state of the art. This would require a systematic review of the literature dealing with stakeholder and public participation in water resource management and more specifically in groundwater management. For example, it would be necessary to report on how conflicting goals and interests are addressed and, if necessary, mitigated through stakeholder participation in other studies. In particular, this section would need to make clear what innovative contribution the present study makes compared to the state of the art.
